# Computational Fluid Dynamics Simulation of Combustion and Selective Non-Catalytic Reduction in a 750 t/d Waste Incinerator

Hai Cao [1,2], Yan Jin [1], Xiangnan Song [3], Ziming Wang [3], Baoxuan Liu [3] and Yuxin Wu [2,*]

1 College of Electrical and Power Engineering, Taiyuan University of Technology, Taiyuan 030024, China
2 Key Laboratory for Thermal Science and Power Engineering of Ministry of Education, Department of Energy and Power Engineering, Tsinghua University, Beijing 100084, China
3 CUCDE Enviromental Technology Co., Ltd., Beijing 100120, China
* Correspondence: wuyx09@mail.tsinghua.edu.cn

**Abstract:** In this study, a Computational Fluid Dynamics (CFD) approach using Ansys Fluent 15.0 and FLIC software was employed to simulate the combustion process of a 750 t/d grate-type waste incinerator. The objective was to assess the performance of Selective Non-Catalytic Reduction (SNCR) technology in reducing nitrogen oxide ($NO_x$) emissions. Two-stage simulations were conducted, predicting waste combustion on the bed and volatile matter combustion in the furnace. The results effectively depicted the temperature and gas concentration distributions on the bed surface, along with the temperature, velocity, and composition distributions in the furnace. Comparison with field data validated the numerical model. The findings serve as a reference for optimizing large-scale incinerator operation and parameter design through CFD simulation.

**Keywords:** grate furnace; SNCR; gas-solid two-phase combustion coupling; numerical simulation

## 1. Introduction

Waste recycling has emerged as a prominent global issue. The generation of solid waste, especially municipal solid waste, has been steadily increasing due to population growth and economic prosperity. This surge in waste production has raised substantial concerns related to the environment and public health. Consequently, regions worldwide are proactively pursuing urban solid waste management strategies to promote sustainable development and effectively tackle these urgent environmental challenges [1]. Waste incineration is widely used as a waste treatment method for power generation because it has the characteristics of high waste reduction, low pollution, and high conversion rate and is expected to become the main form of waste treatment in the future [2]. Waste incineration is carried out in waste incinerators, and usually, incineration power plants are built in urban areas, so regions have strict requirements for pollutant emissions from waste incinerators, and controlling the emissions of various types of typical pollutants has become increasingly important [3].

Compared with other waste treatment technologies, the grate-type waste incinerator is of great significance [4]. However, the secondary pollution problem still exists for incineration, which is mainly harmful gases generated during incineration, such as $NO_x$. Such pollutants can pose a great threat to the environment and human health, such as acid rain and photochemical smog [5]. Given these considerations, numerous researchers have directed their attention toward mitigating nitrogen oxides. Among the approaches for nitrogen oxide removal, two prevalent methods are staged combustion and Selective Non-Catalytic Reduction (SNCR), with SNCR technology enjoying broader applications [6,7]. Pre-combustion control is also called primary control, which mainly prevents the production of nitrogen oxides and post-combustion control is also called secondary control, which mainly reduces nitrogen oxides after they are generated. Of the two $NO_x$ treatment methods, the Selective Non-Catalytic Reduction (SNCR) technology has

been widely used for post-combustion $NO_x$ control methods because of its high economy, simple operation, catalyst-free system, and easy installation [8]. While in the past decade, SNCR technology has widely been used in coal-fired boilers to reduce $NO_x$ emissions, the reductant is denitrified by spraying into the upper part of the furnace chamber at high temperatures [5].

Despite the many advantages of this technology as described above, very severe operating conditions are required during actual operation. The efficiency of SNCR is strongly influenced by many factors, including: the appropriate temperature window, the appropriate flue gas residence time, the ammonia-nitrogen molar ratio, and perturbations [9]. Usually, the generation of $NO_x$ in the flue gas of a waste incinerator is mainly caused by the oxidation of the nitrogenous components of the waste.

Due to the actual operation of the incinerator site in the process of parameter measurement being difficult, time-consuming, and dangerous, especially for the normal operation of SNCR technology, while needing detailed data to carry out the correct guidance in order to accurately and qualitatively quantify the SNCR process, an in-depth understanding of the heat and mass transfer in the furnace, the flow situation, and the reaction kinetics are indispensable. Computational Fluid Dynamics (CFD), in contrast, has many advantages, such as low cost, maneuverability, and the ability to effectively observe the combustion process, which has become an indispensable tool for more and more researchers to study the details of combustion inside the grate furnace [8]. Since combustion in a grate furnace can be divided into two parts: solid-phase combustion in the bed and gas-phase combustion in the bed, the research methods usually have two kinds of combustion studied separately and as a whole. The first research method is to simulate the solid-phase combustion in the bed and the gas-phase combustion in the bed. In this method, the first step is to make use of known fuel information and typical operating parameters that need to be calculated, such as industrial analysis, elemental analysis and calorific value of the fuel. For the typical operating parameters of the grate furnace to be calculated, mainly include the grate size, primary air velocity, and temperature, etc. Then, the calculation results are obtained through the first step of the calculation as the input information as the CFD software (Ansys Fluent 15.0) for the calculation of the initial conditions for the calculation; the results of the two are iteratively coupled to each other until convergence [10]. Another research method is to combine fluids and solids and consider the bed as a porous medium so that the bed solid-phase combustion can also be divided into the entire computational domain, and many processes of solid-phase combustion take place in this region. The computational difficulty of this method, although more difficult than the first method, is characterized by the ability to directly observe the process of change in the interaction between the gas and solid phases. For instance, Lin et al. introduced a fluid-structure coupling mechanical model that relies on the Rankine vortex model and Helmholtz equation to determine critical penetration conditions. This model presents a technical solution that holds great potential for industrial monitoring systems [11]. Chen and colleagues have conducted research on the material transport process in gas-liquid-solid three-phase mixed flows, building upon the coupling of gas-solid two-phase systems. They proposed a coupled modeling and solving approach based on the soft-sphere and porous media models, combining Computational Fluid Dynamics (CFD) and the discrete element method (DEM). This research explores the transport mechanisms of materials and holds significant importance in enhancing the quality and efficiency of material mixing [12].

In this paper, the converged combustion results are used as a benchmark for SNCR denitrification calculations.

The waste enters the incinerator through the feeding port, is pushed onto the incinerator grate by the pusher, and is first heated by the radiation influence of the flame and the wall surface from the upper part of the bed layer, and the temperature is gradually transmitted downward to the grate along the bed top; meanwhile, the primary air supplied at the bottom of the grate carries out convection heat exchange with the waste, so as to realize the rapid evaporation of water in the waste. After the moisture removal, the waste

continues to absorb heat until the combustible material reacts violently with oxygen, which is the highest temperature area of the bed. The waste will be pyrolyzed when it absorbs heat, and the volatilized products will enter the furnace for continuous combustion, and the ash generated after incineration will be discharged through the tail slag outlet.

## 2. Models and Simulation Method

### 2.1. Parameters of the Waste Incinerator

In this paper, a 750 t/d municipal solid waste incinerator with a rated steam pressure of 4.0 Mpa and rated steam temperature of 673 K is used. Due to the complex structure of the actual incinerator, the furnace body is appropriately simplified, and its front view, left view, right view, and top view are shown in Figure 1a–d, respectively. The height of the incinerator is 27.557 m, the length of the grate is 13.464 m, the inclination angle is 15°, and the moving speed of the grate is 7.2 m/h. The secondary air and return flue gas nozzles are arranged at the throat of the incinerator, and the quantity is as follows: There are 6 front secondary air nozzles, 12 front backflow flue gas nozzles, 7 rear secondary air nozzles, and 13 rear backflow flue gas nozzles. The secondary air nozzles and backflow flue gas nozzles are arranged in a staggered manner. The urea nozzle is arranged in the area above the waste heat boiler.

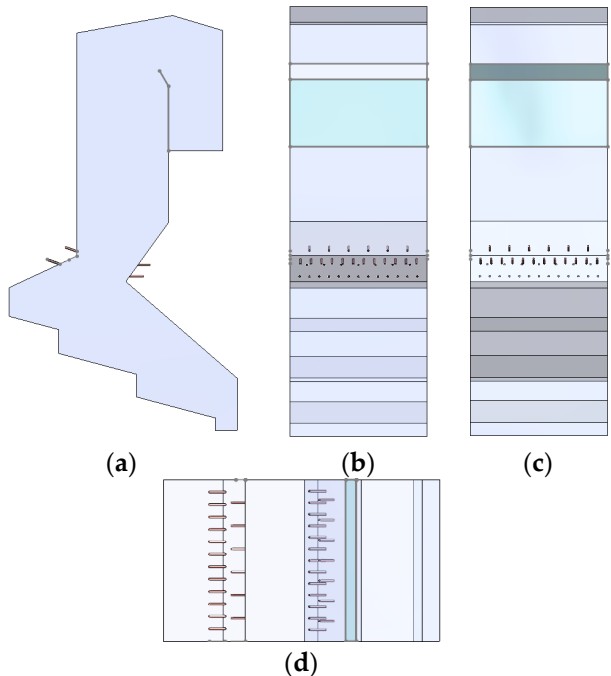

**Figure 1.** Geometric view of incinerator. (**a**) front view, (**b**) left view, (**c**) right view, and (**d**) top view.

In order to improve the accuracy of the numerical simulation, compared with the conventional tetrahedral mesh, the hexahedral mesh is used for the whole calculation domain in the model, as shown in Figure 2a. In view of the large fluid velocity gradient change and complex flow field in the area above the grate and the secondary air and reflux flue gas outlet, the grid in this area is appropriately densified and gradually transited to the upper flue of the incinerator, and the classical O-type division is adopted for the circular tube, which can not only capture more detailed data but also improve the grid quality and save computing resources. The grid of the incinerator throat area and single nozzle grid are shown in Figure 2b,c. The number of grids is 1,847,019, and the grid independence has been verified before calculation. Grids with this number have good performance in velocity, temperature, concentration, and calculation time.

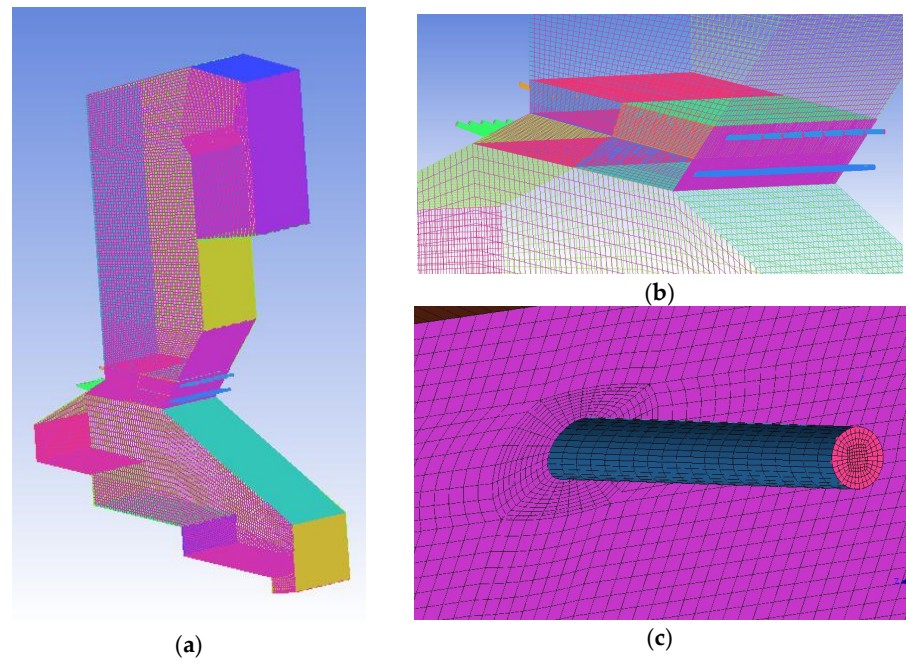

**Figure 2.** Schematic diagram of incinerator model meshing. (**a**) Overall grid; (**b**) Grid zoning near nozzle group; (**c**) Local mesh of single nozzle.

The air is proportionally divided into primary air and secondary air. The primary air is blown in from the bottom of the grate, which can not only make the moisture in the garbage evaporate by heat absorption but also provide sufficient oxygen to provide favorable conditions for the full incineration of the garbage. The secondary air enters the incinerator through the nozzle set at the throat of the incinerator to provide sufficient oxygen for the secondary combustion of combustible gases in the furnace, to provide perturbation, strengthen heat transfer, and accelerate the reaction. In this paper, the ratio of primary air and secondary air is 0.978:0.022, and there are six air boxes on the surface of the grate for the supply of primary air, and the temperature of the primary air is 433.15 K. The analysis of municipal solid waste reveals a consistently high moisture content, as indicated in Table 1. Table 2 provides the industrial and elemental analysis of the studied garbage, as detailed in this paper.

**Table 1.** Element analysis of garbage industry analysis in other studies [2].

| Ultimate Analysis (wt%) | MSW | Proximate Analysis (wt%) | MSW |
|---|---|---|---|
| Carbon | 56.9 | Moisture | 45.32 |
| Hydrogen | 8.75 | Volatile | 25.36 |
| Oxygen | 32.37 | Fixed carbon | 11.46 |
| Nitrogen | 0.38 | Ash | 17.86 |
| Sulfur | 0.41 | LHVar | 7.98 MJ/kg |

**Table 2.** Proximate analysis and ultimate analysis (day ash-free basis) of the waste.

| Ultimate Analysis (wt%) | Mcr Case | Proximate Analysis (wt%) | MCR Case |
|---|---|---|---|
| Carbon | 55.39 | Moisture | 49.2 |
| Hydrogen | 10.98 | Volatile | 27.16 |
| Oxygen | 30.59 | Fixed carbon | 5.43 |
| Nitrogen | 1.66 | Ash | 18.21 |
| Sulfur | 1.35 | LHVar | 7536 kJ/kg |

### 2.2. Simulation Method

The current mainstream numerical simulation for waste incineration according to different mathematical models and computational procedures can be divided into two parts of the overall combustion simulation of waste in the incinerator, respectively, the use of the FLIC program to simulate the combustion of waste on the grate (two-dimensional simulation) and commercial CFD software (Ansys Fluent 15.0) FLUENT for the simulation of combustion in the furnace (three-dimensional simulation), which are iteratively coupled [13,14].

The FLIC (Fluid dynamic Incinerator Code) program was used for the simulation of grate bed combustion. The waste undergoes heat and mass exchange between solid and gas phases on the grate. In this software, the bed simulation consists of the following four steps, namely, moisture evaporation, de-volatilization, volatile combustion, and coke combustion, which covers complex physicochemical processes, such as heat conduction, heat convection, heat radiation, and mass transfer, and the four processes are interrelated and interact with each other with no obvious demarcation [15]. The temperature distribution, component distribution, and velocity distribution of the gas and solid in the bed can be obtained by FLIC software calculation.

The FLIC software, developed by the University of Sheffield in the United Kingdom, is utilized for calculating solid-phase combustion in the bed layer. When using FLIC, the essential inputs include the geometric dimensions of the grate, operating parameters simulated conditions, and the industrial and elemental analysis of the waste. It is necessary to set discrete grid cells. By solving the mathematical equations for solid waste combustion using the software, one can obtain the distribution of the temperature, velocity, and composition of volatile matter generated in the bed layer during solid waste combustion.

Please refer to the accompanying flowchart for a basic illustration of the FLIC method in Figure 3.

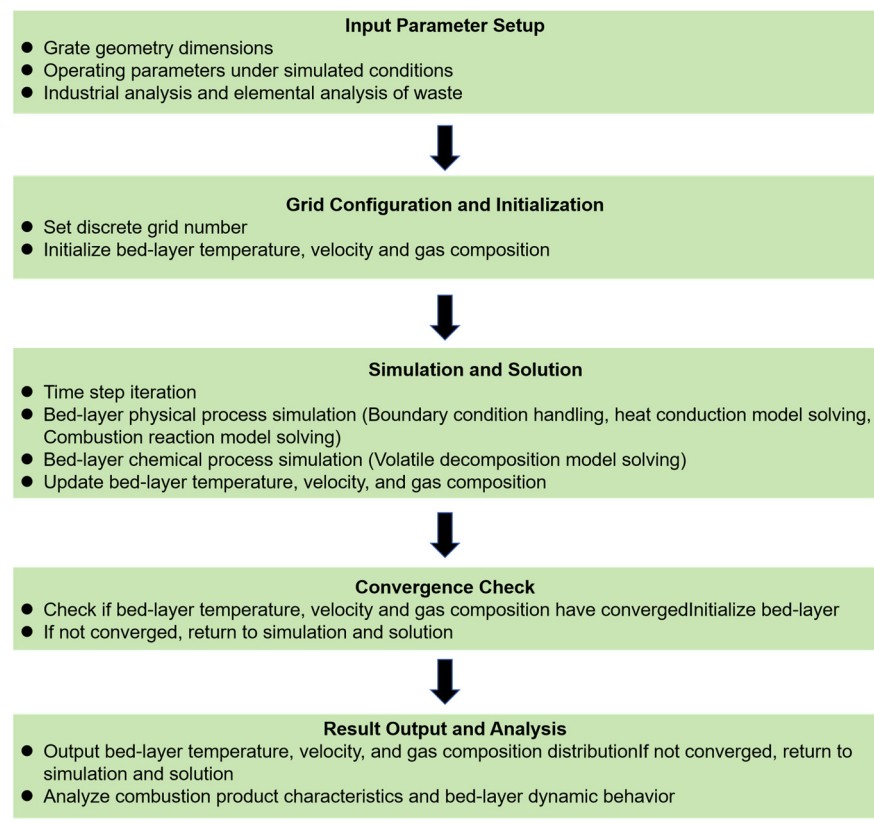

**Figure 3.** The basic flow of the FLIC simulation calculation.

FLUENT is used to solve the problem of turbulent mixing as well as secondary combustion of volatile fractions. The volatile components from the waste pyrolysis continue to be combusted in the furnace. The simulation inside the furnace requires the temperature distribution, velocity distribution, and component distribution provided by FLIC, as described above. At the same time, the FLUENT simulation also provides FLIC with the radiative heat flow density along the grate's length. The two are coupled through iterations until the radiant heat flow density and gas temperature end without significant change with an increasing number of iterations [16].

A schematic diagram of the incinerator numerical simulation method is shown in Figure 4.

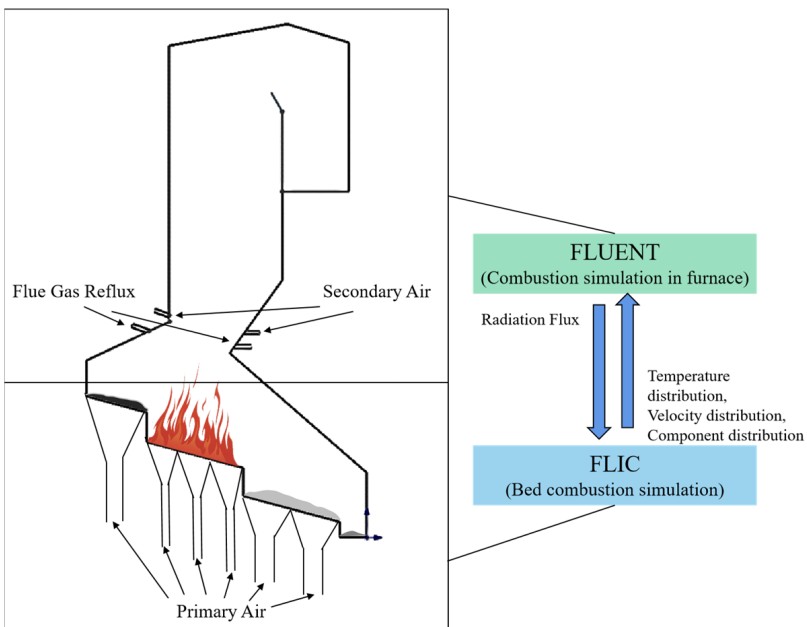

**Figure 4.** Numerical simulation methods for incinerators.

### 2.2.1. Bed Combustion Simulation

For bed combustion simulation, the temperature distribution and reaction process can be obtained by solving the two-dimensional conservation equation of the gas-solid two-phase. Because municipal solid waste has a wide range of sources, diversity of composition, uncertainty of calorific value, combustion instability, and complex combustion process, it is difficult to calculate. Therefore, appropriate simplification and assumptions need to be made: (1) the heterogeneity caused by the bed along the width direction is ignored. The bed calculation can only solve the two-dimensional case. (2) The moving velocity of the bed is constant. (3) The solid phase contains only moisture, volatile matter, fixed carbon, and ash. (4) Waste can be regarded as equal-diameter spherical particles during combustion on the grate. (5) The bed is regarded as a continuous porous medium [17].

The basic conservation equation of the solid phase in the bed is as follows [2]:

Continuity:

$$\frac{\partial \rho_{sb}}{\partial t} + U_b \frac{\partial \rho_{sb}}{\partial x} + \frac{\partial(\rho_{sb}V_s)}{\partial y} = -S_s \tag{1}$$

Energy:

$$\frac{\partial(\rho_{sb}H_s)}{\partial t} + U_b \frac{\partial(\rho_{sb}H_s)}{\partial x} + \frac{\partial(\rho_{sb}H_sV_s)}{\partial y}$$
$$= \frac{\partial(\lambda_s \frac{\partial T_s}{\partial x})}{\partial x} + \frac{\partial(\lambda_s \frac{\partial T_s}{\partial y})}{\partial y} - S_a h_s'(T_s - T_g) + \sum \Delta h_k \tag{2}$$

Species:

$$\frac{\partial(\rho_{sb}Y_{is})}{\partial t} + U_b\frac{\partial(\rho_{sb}Y_{is})}{\partial x} + \frac{\partial(\rho_{sb}Y_{is}V_s)}{\partial y}$$
$$= \frac{\partial}{\partial x}\left[D_s\frac{\partial(\rho_{sb}Y_{is})}{\partial x}\right] + \frac{\partial}{\partial y}\left[D_s\frac{\partial(\rho_{sb}Y_{is})}{\partial y}\right] - S_{is} \tag{3}$$

The solid phase velocity adopts the grate moving speed, which is 0.002 m/s. The gas phase equation in the bed is similar to the solid phase equation.

Continuity:

$$\frac{\partial(\phi\rho_g)}{\partial t} + \frac{\partial(\phi\rho_gU_g)}{\partial x} + \frac{\partial(\phi\rho_gV_g)}{\partial y} = S_s \tag{4}$$

X-Momentum:

$$\frac{\partial(\phi\rho_gU_g)}{\partial t} + \frac{\partial(\phi\rho_gU_gU_g)}{\partial x} + \frac{\partial(\phi\rho_gV_gU_g)}{\partial y} = -\frac{\partial p_g}{\partial x} + F(U_g) \tag{5}$$

Y-Momentum:

$$\frac{\partial(\phi\rho_gU_g)}{\partial t} + \frac{\partial(\phi\rho_gU_gV_g)}{\partial x} + \frac{\partial(\phi\rho_gV_gU_g)}{\partial y} = -\frac{\partial p_g}{\partial y} + F(V_g) \tag{6}$$

Energy:

$$\frac{\partial(\phi\rho_gH_g)}{\partial t} + \frac{\partial(\phi\rho_gH_gU_g)}{\partial x} + \frac{\partial(\phi\rho_gH_gV_g)}{\partial y} =$$
$$\frac{\partial(\lambda_g\frac{\partial T_g}{\partial x})}{\partial x} + \frac{\partial(\lambda_g\frac{\partial T_g}{\partial y})}{\partial y} + S_ah'_s(T_s - T_g) + \sum\Delta h_k \tag{7}$$

Species:

$$\frac{\partial(\phi\rho_gY_{jg})}{\partial t} + \frac{\partial(\phi\rho_gY_{jg}U_g)}{\partial x} + \frac{\partial(\phi\rho_gY_{jg}V_g)}{\partial y} =$$
$$\frac{\partial}{\partial x}\left[D_g\frac{\partial(\phi\rho_gY_{jg})}{\partial x}\right] + \frac{\partial}{\partial y}\left[D_g\frac{\partial(\phi\rho_gY_{jg})}{\partial y}\right] + S_{js+}S_{jg} \tag{8}$$

The laminar combustion of MSW on the grate can be divided into the following four processes: water evaporation, volatilization analysis, volatile combustion, and char combustion. It is important to note that these processes are not distinctly separated and can overlap. Sometimes, they can occur simultaneously. The corresponding bed combustion process diagram is given in Figure 5, followed by the relevant reaction formulas. In this section, the particle characteristics assume that the waste consists of equal-diameter spherical particles with a diameter of 5 mm for simulation purposes.

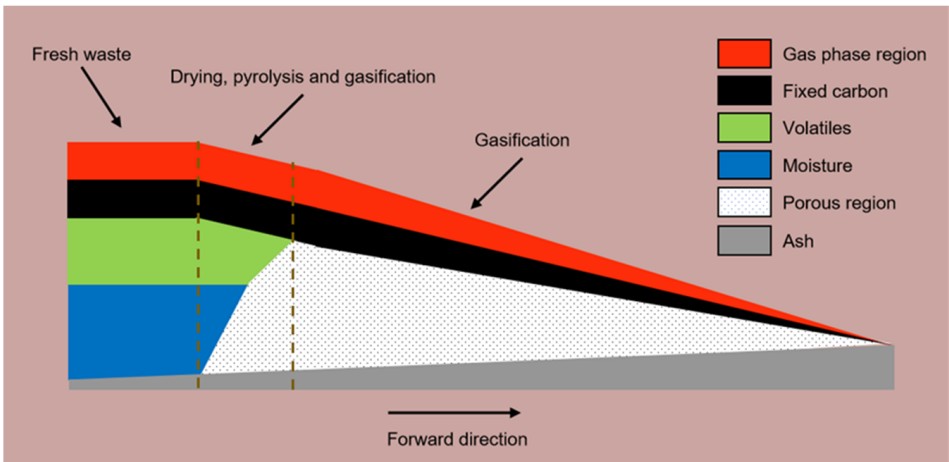

**Figure 5.** Bed-layer combustion process.

Water Evaporation

After fresh waste is fed into the grate, the water within the waste begins to evaporate due to the combined effect of radiation from the furnace and convection from the primary air. In this paper, it is assumed that the evaporation of water commences once the temperature reaches 373 K. The specific expression for this phenomenon is as follows:

$$
R_M = \begin{cases} A_s h_s (C_{w,s} - C_{w,g}) & T_s < 373 \cap \rho_M > 0 \\ \dfrac{A_s [h'(T_g - T_s) + \varepsilon_s \sigma_b (T_{env}^4 - T_s^4)]}{H_{evap}} & T_s = 373 \cap \rho_M > 0 \\ 0 & T_s > 373 \cup \rho_M > 0 \end{cases} \tag{9}
$$

Volatilization Analysis

When the surface temperature of the waste reaches a specific threshold, volatiles start to precipitate from the waste particles. A correlation exists between the volatilization rate and the residue of volatile matter. The specific expression describing this association is as follows:

$$
R_{vol} = k_{vol} \rho_{vol} (C_{vol,s} - C_{vol,g}) \tag{10}
$$

$$
k_{vol} = A_{vol} \exp(-\frac{E_{vol}}{RT_s}) \tag{11}
$$

Volatile Combustion

After the volatilization analysis, the volatiles are initially mixed with the surrounding air before undergoing combustion. The combustion rate is influenced by both the kinetic reaction rate and the mixing rate. The specific expression of the combustion rate is as follows:

$$
R = \min(R_{kin}, R_{mix}) \tag{12}
$$

$$
R_{mix} = C_{mix} \left[ 150 \frac{D_g (1 - \phi)^{\frac{2}{3}}}{d_P^2 \phi} + 1.75 \frac{V_g (1 - \phi)^{\frac{1}{3}}}{d_p \phi} \right] \min(\frac{C_{fuel}}{S_{fuel}}, \frac{C_{o_2}}{S_{o_2}}) \tag{13}
$$

Char Combustion

Char combustion represents the final stage of the bed combustion process. The primary combustion products derived from char include carbon monoxide (CO) and carbon dioxide ($CO_2$). The specific expression for this phenomenon is as follows:

$$
C_{(s)} + \alpha O_2 \rightarrow 2(1 - \alpha)CO + (2\alpha - 1)CO_2 \tag{14}
$$

$\alpha$ represents the oxygen consumed by unit char combustion. The ratio of CO to $CO_2$ is:

$$
\frac{CO}{CO_2} = 2500 \exp(\frac{-6420}{T}), \ 730 \text{ K } < \text{ T} < 1170 \text{ K} \tag{15}
$$

2.2.2. Combustion Simulation in Furnace

The continuity equation, energy equation, component equation, and momentum equation of furnace combustion are similar to those of in-bed combustion. Details can be found in the FLUENT manual [18].

The governing equation of gaseous combustion simulation is solved by the SIMPLE algorithm, the interaction between turbulence and chemical substances is solved by the finite rate-eddy dissipation model, the turbulence model is based on the standard k-ε model [19], and the radiative heat transfer is based on the P-1 model [20]. Considering the buoyancy effect of actual flow in the furnace, the gravity in the whole calculation domain is opened up. The model and solution selected in the furnace simulation are shown in Table 3.

**Table 3.** The model selected in the furnace simulation and its solution.

| Combustion Model in the Furnace | | Solution Method |
|---|---|---|
| Turbulence | Realizable k-ε | SIMPLE |
| Radiation | P-1 | |
| Interaction between turbulence and chemicals | FR/ED | Second-order upwind |

In the calculation, the boundary conditions are considered to be as follows: the water wall is approximately treated as the constant wall temperature boundary condition, the drop between the grate and the grate adopts the adiabatic boundary condition, the third section grate, the secondary air, and the reflux flue gas all adopt the velocity inlet boundary condition, and the flue gas outlet is the pressure outlet.

In order to simplify the model and speed up the calculation, the $C_mH_n$ in the flue gas only considers methane, and the other components are $O_2$, $CO$, $H_2O$, $CO_2$, and $H_2$ [21]. The reaction equation in the furnace is as follows [22]:

$$CH_4 + 1.5O_2 \rightarrow CO + 2H_2O \tag{16}$$

$$CO + 0.5O_2 \rightarrow CO_2 \tag{17}$$

$$H_2 + 0.5O_2 \rightarrow H_2O \tag{18}$$

The corresponding expression of the chemical reaction rate is as follows [22]:

$$k_1 = A_1 T^{\beta_1} \exp(-\frac{E_1}{RT}) \cdot [CH_4]^{0.7}[O_2]^{0.8} \tag{19}$$

$$k_2 = A_2 T^{\beta_2} \exp(-\frac{E_2}{RT}) \cdot [CO][O_2]^{0.25} \tag{20}$$

$$k_3 = A_3 T^{\beta_3} \exp(-\frac{E_3}{RT}) \cdot [H_2][O_2]^{0.5} \tag{21}$$

where $A_1 = 5.012 \times 10^{11}$, $\beta_1 = 0$, $E_1 = 2.0 \times 10^8$, $A_2 = 2.239 \times 10^{12}$, $\beta_2 = 0$, $E_2 = 1.7 \times 10^8$, $A_3 = 9.870 \times 10^8$, $\beta_3 = 0$, $E_3 = 3.1 \times 10^7$.

## 3. Results

### 3.1. Simulation Results of Bed Combustion

The simulation in this paper is divided into bed combustion simulation and hearth combustion simulation. In the bed combustion simulation, the pre-assumed radiation temperature is used for the in-bed calculation. Then the gas component distribution, gas temperature distribution, and gas velocity distribution along the length of the grate are obtained. The above three distributions are used as the inlet boundary conditions of the grate in the hearth combustion simulation, and the radiation temperature on the grate is obtained after the hearth simulation, and then this radiation temperature is returned to the bed combustion simulation, and the two are iterated continuously until the gas temperature and the density of the radiant heat flux no longer change.

The gas temperature at the bed surface along the length of the grate is given as shown in Figure 6, where the total length of the grate is 13.464 m. The waste drying time is longer because of the high moisture content of the waste, i.e., the drying section of the incinerator occupies a larger part of the grate. At about 5.5 m along the grate length, the flue gas temperature rises abruptly to about 1050 K. As the garbage moves on the grate, the temperature of the flue gas at about 5.8 m will drop to about 330 K, and then the volatile matter will start to precipitate rapidly and catch fire, i.e., it will enter the pyrolysis stage, and the temperature will climb rapidly between 5.9 m and 6.5 m to 1430 K. As the combustion

proceeds, the temperature will be reduced gradually, and when the coke precipitates and continues to be combusted, the temperature will rise to the highest temperature of 1520 K. Finally, the temperature will be reduced by the amount of the coke, and the fixed carbon, which is the largest part of the drying section of the incinerator. With the exhaustion of volatile components and fixed carbon, the flue gas temperature on the bed surface gradually decreases and stabilizes at about 510 K.

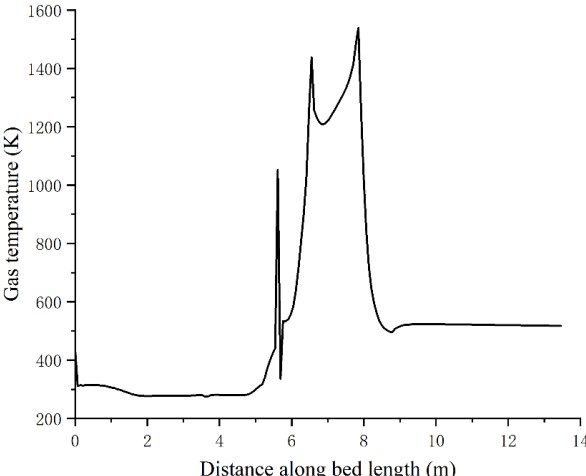

**Figure 6.** Bed gas temperature along the grate length.

The gas-phase component distribution on the bed surface along the length of the grate and the bed combustion process curves are given, as shown in Figures 7 and 8, respectively. The water precipitates quickly after the garbage is fed into the furnace, and the fresh garbage loses weight rapidly, and the weight loss rate of the garbage is as high as 81.74%. Since the moisture is removed from the garbage during the drying process without combustion, there is almost no oxygen consumption at this time, so there is almost no need for oxygen supply before 3.6 m. Due to the high water content, the evaporation of water is basically completed at about 5.2 m. The pyrolysis process lasts from about 5.4 m to 8 m, the volatile matter is precipitated in this stage, so the content of $C_mH_n$ and CO is higher in this stage. The ignition and combustion of the volatile components rapidly reduced the $O_2$ content to nearly 0, and then entered the stable combustion stage, so the $O_2$ content gradually recovered to about 23% until 8 m. In addition, the $CO_2$ in the flue gas at the bed surface mainly originated from the precipitation and combustion process of volatile matter, while the content in the drying and burnout sections was very small.

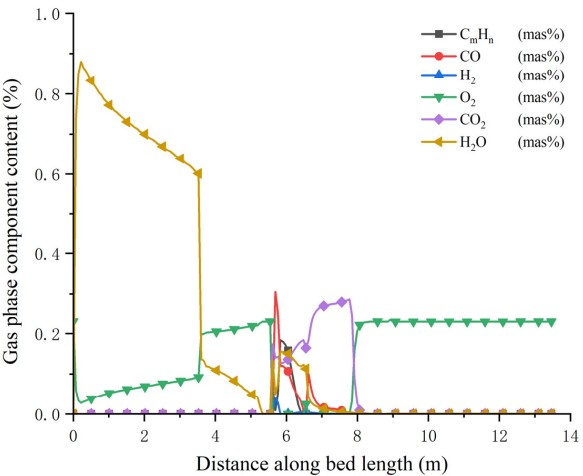

**Figure 7.** Variation curve of the gas phase mass fraction.

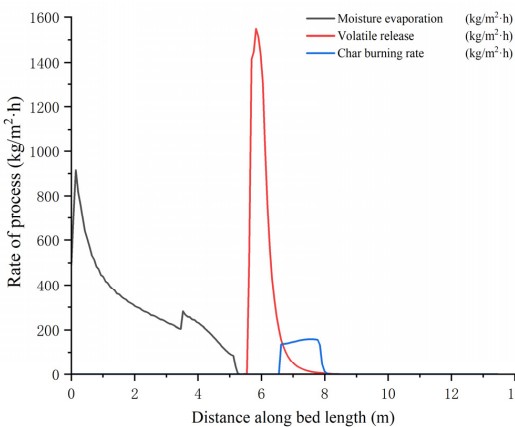

**Figure 8.** Bed combustion process curve.

### 3.2. Simulation Results of Gas Combustion

The combustion in a municipal solid waste incinerator was simulated under MCR conditions. The primary air blowing from the bottom grate will carry the combustible gas from garbage pyrolysis into the furnace to continue to burn, and the secondary air from the front wall and back wall will provide enough oxygen for the unburned gas to ensure full combustion. The temperature distribution in the central section of the combustion chamber is shown in Figure 9a, and the temperature distributions near the secondary air and reflux flue gas nozzles are shown in Figure 9b,c,d and e, respectively. It can be seen that the highest temperature of the bed is 1400 K, which appears in the second section of the grate, that is, the grate in the combustion section. This is because the garbage begins to burn and the volatilization is analyzed in this area, and the combustion flame appears in the bed. As the flue gas continues to flow upward, the flue gas temperature will decrease due to the heat absorption of the water wall. However, with the secondary air blowing into the front and rear wall and the reflux flue gas, the unburned gas reacts with oxygen again, and the disturbance in this area is strong, so a large amount of heat will be released, and the flue gas temperature will rise again along the secondary air path, which is 1600 K. However, the temperature decreases gradually from the end of the second stage grate to the third stage grate because the fixed carbon is basically consumed, the combustible gas content is low, and the combustion is difficult to maintain.

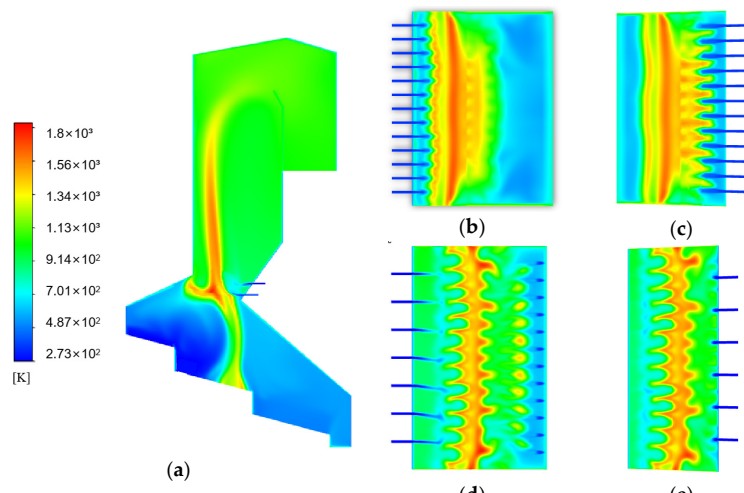

**Figure 9.** Temperature distribution in the furnace. (**a**) Temperature distribution of central section of furnace. (**b**) Temperature distribution of reflux flue gas (left). (**c**) Temperature distribution of reflux flue gas (right). (**d**) Secondary air temperature distribution (left). (**e**) Secondary air temperature distribution (right).

### 3.3. Validating Model Accuracy

To ensure the precision and reliability of the chosen model and simulation strategy, a comparison is drawn between the simulated combustion results and actual field operation measurements. The spatial arrangement of temperature measuring points in the field is illustrated in Figure 10. Recognizing the temporal variations in measuring point values, the daily average temperature is utilized as a substitute for individual temperature measurements. This approach enables a robust assessment of simulation effectiveness by aligning it with real-world operational data.

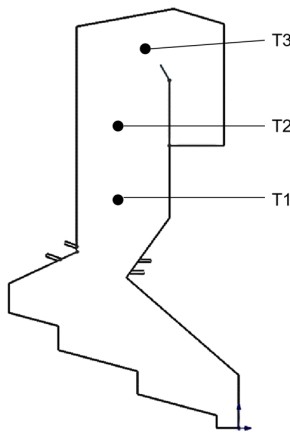

**Figure 10.** Schematic diagram of the temperature measuring point. T1–T3 is the furnace entrance, the middle of the furnace and the furnace outlet.

Table 4 displays the comparison between temperature measurements from the measuring points and the corresponding simulation outcomes.

**Table 4.** Comparison between the temperature value of the temperature measuring point and the simulation results.

| Point | Item | Simulation Results (K) | Actual Result (K) | Error (%) |
|-------|------|------------------------|-------------------|-----------|
| T1 | Flue gas temperature in the lower part of the furnace | 1151.02 | 1162.43 | 0.98 |
| T2 | Flue gas temperature in the middle of the furnace | 990.69 | 972.07 | 1.92 |
| T3 | Flue gas temperature at the top of the furnace | 1021.33 | 1031.90 | 1.31 |

The comparison table highlights a close alignment between the simulated temperature distribution and the real operational data, with an approximate 1% margin of error. Overall, the model and methodology employed in this study exhibit effectiveness and rationality in simulating combustion outcomes, thus providing a solid groundwork for subsequent SNCR investigations.

### 3.4. $NO_x$ Distribution in the Furnace without SNCR Addition

The nitrogen oxides produced during combustion can be divided into three types: thermal nitrogen oxides, fuel nitrogen oxides, and fast nitrogen oxides. There are corresponding models for solving these three types of nitrogen oxides in FLUENT.

The calculated $NO_x$ temperature distribution in the central section of the furnace is shown in Figure 11. The formation temperature of thermal $NO_x$ is about 1800 K, so there is almost no thermal $NO_x$ in the process of MSW incineration. The $NO_x$ in the flue gas mainly comes from the oxidation of trace nitrogen compounds in MSW during combustion, and the temperature has little effect on fuel-type $NO_x$, so fuel-type $NO_x$ is the main one. It can be seen from the figure that the concentration of $NO_x$ near the entrance of the secondary tuyere is obviously higher than that of other positions, which is mainly due to the fact that the secondary air provides sufficient oxygen for the formation of $NO_x$ and accelerates

the formation of $NO_x$ under severe disturbances. The calculated result of the average concentration of $NO_x$ at the exit section is 2.658 ppm.

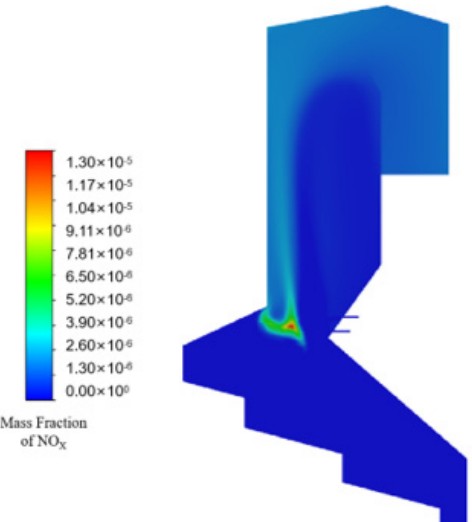

**Figure 11.** $NO_x$ distribution in the furnace without SNCR.

In Beijing, China, there are stringent regulations regarding the maximum permissible concentration of $NO_x$ in the exhaust, which should not exceed 14.63 ppm. Based on the original emission data presented in this paper, the concentration of $NO_x$ is measured at 2.658 ppm, which is lower than the specified limit.

*3.5. $NO_x$ Distribution in Furnace after Using SNCR*

In the actual production practice, the most commonly used amino-reducing agents are ammonia and urea. From the point of view of safety, urea is selected as the reducing agent for calculation. Urea is first pyrolyzed to produce ammonia, which then reacts with NO.

The opening surface of the boiler is the front wall of the first flue and the left and right side walls, and the front wall of the first flue has four holes in each layer, with a total of three layers. The sidewall of the first flue has three holes in each layer, with a total of three layers. The $NO_x$ distribution after the nitrogen oxide treatment with urea is shown in Figure 12, so it is not difficult to see that the proportion of pollutants obviously decreases.

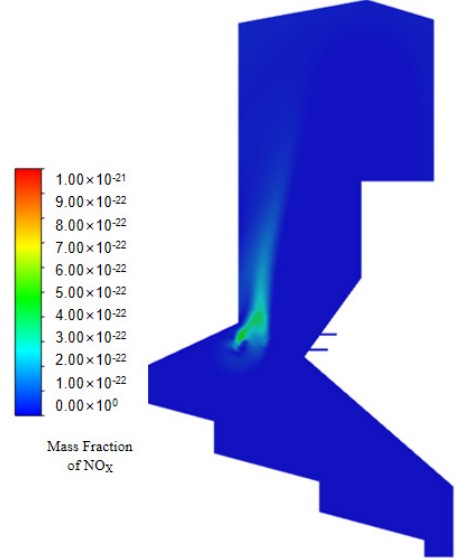

**Figure 12.** $NO_x$ distribution in the furnace after using SNCR.

## 4. Conclusions

The main objective of this study was to investigate the combustion characteristics of a municipal solid waste incinerator under Maximum Continuous Rating (MCR) conditions and perform numerical simulations of Selective Non-Catalytic Reduction (SNCR). This research is significant as waste incinerators play a crucial role in urban waste treatment, and understanding their combustion characteristics and methods for reducing nitrogen oxide emissions can enhance furnace efficiency and environmental friendliness.

To comprehensively simulate the waste combustion process inside the furnace under MCR conditions, we employed an effective numerical simulation method. By accurately establishing the model and setting appropriate parameters, we ensured the accuracy of the simulation results and controlled errors within a reasonable range. Notably, we provided a detailed description of the methodology for studying the temperature field inside the furnace using FLIC-FLUENT software, offering readers clear operational guidelines and practical techniques.

Building upon the study of the temperature field inside the furnace, we further conducted FLUENT calculations to understand the distribution of nitrogen oxides ($NO_x$) within the furnace. Although the initial emission concentrations did not exceed stringent limits under harsh conditions, our subsequent SNCR research demonstrated the effectiveness of Selective Non-Catalytic Reduction technology in reducing nitrogen oxide emission concentrations. These findings validate the practicality and feasibility of the research model and methodology employed in this paper, providing strong support for future waste incinerator studies.

In summary, this study delved into the combustion characteristics of waste incinerators under MCR conditions and explored the potential application of SNCR technology in reducing nitrogen oxide emissions through numerical simulations. These research findings establish a solid foundation for obtaining the temperature field inside waste incinerators and performing numerical simulations of SNCR. Furthermore, they provide valuable references for improving the operational efficiency of waste incinerators and reducing environmental pollution.

**Author Contributions:** Conceptualization, Y.W.; Methodology, H.C.; Software, X.S., Z.W. and B.L.; Investigation, H.C.; Writing—original draft, H.C.; Writing—review & editing, H.C.; Visualization, H.C.; Supervision, Y.W.; Project administration, Y.J. and Y.W. All authors have read and agreed to the published version of the manuscript.

**Funding:** This work was supported by the Key Project of Natural Science Foundation of China (No. 29936090), the Special Funds for Major State Basic Research projects (G1999022102), Huaneng Group Science and Technology Research Project (HNKJ22-H105) and Seed fund of Shanxi Research Institute for Clean Energy, Tsinghua University.

**Data Availability Statement:** All the data generated or analyzed in this study are included in this published article.

**Conflicts of Interest:** The authors declare no conflict of interest.

## Nomenclature

| | |
|---|---|
| $A$ | predigital factor, $s^{-1}$ |
| $A_s$ | particle surface area, $m^2$ |
| $A_{vol}$ | pre-exponential factor from volatilization analysis, $1/s$ |
| $C_{fuel}$ | volume fraction of combustible gas, % |
| $C_{mix}$ | empirical constant of gas mixing |
| $C_{O2}$ | volume fraction of oxygen, % |
| $C_{vol,s}$ | volatile matter concentration (solid phase), $kg/m^3$ |
| $C_{vol,g}$ | volatile matter concentration (gas phase), $kg/m^3$ |

| | |
|---|---|
| $C_{w,s}$ | water concentration (solid phase), $kg/m^3$ |
| $C_{w,g}$ | water concentration (gas phase), $kg/m^3$ |
| $D$ | mass diffusion coefficients, $m^2/s$ |
| $D_g$ | gas phase diffusion coefficient, $m^2/s$ |
| $d_p$ | particle diameter, m |
| $E$ | activation energy, J/mol |
| $E_{vol}$ | volatile activation energy, J/mol |
| $H_{evap}$ | latent heat of water evaporation, J/kg |
| $H_g$ | gas phase enthalpy, J/kg |
| $H_s$ | solid phase enthalpy, J/kg |
| $h_s$ | convective mass transfer coefficient, $W/(m^2 \cdot K)$ |
| $h'$ | convective heat transfer coefficient, $W/(m^2 \cdot K)$ |
| $h'_s$ | convective heat transfer coefficient between solid and gas, $W/(m^2 \cdot K)$ |
| $\Delta h_k$ | thermal effects of reactions or $k$th processes, $W/m^3$ |
| $k$ | reaction rate |
| $k_{vol}$ | volatilization analyzed rate constant, 1/s |
| $p_g$ | gas pressure, Pa |
| $R$ | general gas constant, $J/(mol \cdot K)$ |
| $R_{kin}$ | kinetic reaction rate of volatile combustion, $kmol/(m^3 \cdot s)$ |
| $R_M$ | the rate of water evaporation, $kg/(m^3 \cdot s)$ |
| $R_{mix}$ | mixing rate of volatile matter, $kmol/(m^3 \cdot s)$ |
| $R_{vol}$ | volatilization analytical rate, $kg/(m^3 \cdot s)$ |
| $T$ | flame temperature, k |
| $T_{env}$ | radiation temperature in furnace, K |
| $T_g$ | gas temperature, K |
| $T_s$ | solid temperature, K |
| $S_a$ | particle surface area, $m^2/m^3$ |
| $S_{fuel}$ | stoichiometric coefficient of combustible gases |
| $S_{jg}$ | mass source term of gaseous species, $kg/m^3 s$ |
| $S_{js}$ | mass sources due to evaporation, devolatilization and combustion, $kg/m^3 s$ |
| $S_{is}$ | source term, $kg/m^3 s$ |
| $S_{O2}$ | stoichiometric coefficient of oxygen |
| $S_s$ | mass loss rate from solid, $kg\, m^{-3}\, s^{-1}$ |
| $U_b$ | bed movement speed, m/s |
| $U_g$ | gas velocity in the direction of the parallel grate component, m/s |
| $V_s$ | perpendicular to bed movement speed, m/s |
| $V_g$ | gas velocity in the direction of the vertical grate component, m/s |
| $x$ | grate length direction, m |
| $y$ | perpendicular to the length of the grate, m |
| $Y_{is}$ | mass fractions of particle compositions (moisture, volatile, fixed carbon and ash) |
| $Y_{jg}$ | mass fractions for gaseous species |
| $t$ | time, s |
| $\rho_{sb}$ | solid bulk density in the bed, $kg/m^3$ |
| $\rho_g$ | gas density, $kg/m^3$ |
| $\rho_M$ | the density of remaining water in a solid, $kg/m^3$ |
| $\rho_{vol}$ | residual volatile matter density, $kg/m^3$ |
| $\phi$ | bed porosity |
| $\lambda$ | thermal conductivity, $W/(m \cdot K)$ |
| $\varepsilon_s$ | material particle emissivity |
| $\sigma_b$ | Stefan-Boltzmann constant, $5.67 \times 10^{-8}\ W/(m^2 \cdot K^4)$ |

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
