# Peer review of "Computational Fluid Dynamics Simulation of Combustion and Selective Non-Catalytic Reduction in a 750 t/d Waste Incinerator"

_processes, doi:10.3390/pr11092790_

Round 1

Reviewer 1 Report

This paper provides a simulation of waste combustion in a fluidized bed and followed by homogenous SNCR in a furnace space. After reading the manuscript, it seems the real situation is quite different, i.e., the fluidized bed combustion is not provided.

In addition to this problem, the following points are suggested to the authors in revising the manuscript:

[1]. Figure 1 is hard to understand, colored 3D pictures should be provided. .

[2]. Figures 2, 3, and 4 can be combined together, and they should be given in colored form.

[3]. In 2.2 "simulation method", the FLIC method should be introduced in detail, since only a few people are aware of this method. .

[4]. In 2.2.1 "bed combustion simulation", the bed diagram should be given, and the reaction formula for waste decomposition ought to be listed. Besides, the particle property is also required.

[5]. In 3.1"simulation results of bed combustion", the fluidization property of the bed is not given.

[6]. In 3.4, what is the NH3 distribution in the furnace, is its outlet concentration fulfill the environmental restriction?

[7]. Figs. 9-13 can be combined under the same figure caption.

Reviewer 2 Report

The authors Cao et. al explores the stable gas-solid two-phase combustion under typical conditions. This work is meaningful and difficult. However, in the reviewer opinion the paper needs minor revisions to be recommendable for publication.

1. In abstract, the key findings and laws of this paper need to be clarified.

2. In introduction, the recent research background of f CFD numerical topics in recent years should be added. The following papers can provide some reference. 10.1016/j.ymssp.2022.110058.  10.3390/pr11082254

3. In Figure 2, what does the number of the numerical model? The author should provide the independence verification result.

4. What is the turbulence Reynolds number? Need to consider turbulence models?

5. In Figures 10-13, the legend in the figure the same legend as in Figure 9 and is the value the same?

6. The conclusions should be expanded and highlight important findings.

No

Reviewer 3 Report

The paper is very well written and is according to the journal's scope, however, I suggest the following minor corrections before acceptance. 

1. The references provided look old, update the list accordingly from recent literature.

2. None of the equations are cited. See equations 1-8

3. FLIC and FLUENT are used ok? What is their version? Registered or not?

4. Why such a type of geometry is chosen? any reason.

5. Check equation 1, is it written correctly? 

6. Figures need labeling along the y-axis. See Figures 7, 8.

7. Tables need to be properly cited.

8. Equation 2 has a summation "delh_k" term. Please explain this, and how this makes a contribution to the D. equation.

9. See equation 1, explain S_s on the right side, and explain its contribution to mass conservation.

10. I will suggest Ferrofluid treatment with the insertion of an electric field inside a porous cavity considering forced convection for addition in revision.

11. Symbols or units added at the end must have a name as Nomenclature.

Minor check required.

Round 2

Reviewer 1 Report

This paper needs a minor revision on Fig.1,which is not updated after revision.

Author Response

Figure 1 has been colored in the manuscript, and as you said, the visual effect is better than black and white. Please check the manuscript.